
# Assessing the capability of different satellite observing configurations to resolve the distribution of methane emissions at kilometer scales

Alexander J. Turner[1,2], Daniel J. Jacob[2], Joshua Benmergui[2], Jeremy Brandman[3],
Laurent White[3], and Cynthia A. Randles[3]

[1]College of Chemistry/Department of Earth and Planetary Sciences, University of California,
Berkeley, CA, USA.

[2]School of Engineering and Applied Sciences, Harvard University, Cambridge, Massachusetts,
USA.

[3]ExxonMobil Research and Engineering Company, Annandale, NJ, USA.

*Correspondence to:* Alexander J. Turner
(alexjturner@berkeley.edu)

**Abstract.** Anthropogenic methane emissions originate from a large number of fine-scale and of-
ten transient point sources. Satellite observations of atmospheric methane columns are an attractive
approach for monitoring these emissions but have limitations from instrument precision, pixel reso-
lution, and measurement frequency. Dense observations will soon be available in both low Earth and
geostationary orbits, but the extent to which they can provide fine-scale information on methane
sources has yet to be explored. Here we present an observation system simulation experiment
(OSSE) to assess the capabilities of different satellite observing system configurations. We conduct a
1-week WRF-STILT simulation to generate methane column footprints at $1.3 \times 1.3$ km$^2$ spatial reso-
lution and hourly temporal resolution over a $290 \times 235$ km$^2$ domain in the Barnett Shale in Northeast
Texas, a major oil/gas field with a large number of point sources. We sub-sample these footprints
to match the observing characteristics of the recently launched TROPOMI instrument ($7 \times 7$ km$^2$
pixels, 11 ppb precision, daily frequency), the planned GeoCARB instrument ($2.7 \times 3.0$ km$^2$ pixels,
4 ppb precision, nominal twice-daily frequency), and other proposed observing configurations. The
information content of the various observing systems is evaluated using the Fisher information ma-
trix and its eigenvalues. We find that a week of TROPOMI observations should effectively provide
regional ($\sim$100 km) information on temporally invariant emissions but is very limited at finer scales.
GeoCARB should provide 4-37% of the total information available for temporally invariant emis-
sions in the Barnett Shale ($\sim$100 pieces of information). Improvements to the instrument precision
yield greater increases in information content, compared to improved sampling frequency. A preci-
sion better than 6 ppb is an important threshold for achieving fine resolution of emissions. Transient



emissions would be missed with either TROPOMI or GeoCARB. An aspirational high-resolution
geostationary instrument with $1.3 \times 1.3$ km$^2$ pixel resolution, hourly return time, and 1 ppb precision
would effectively constrain the temporally invariant emissions in the Barnett Shale at the kilometer
scale and provide some information on transient sources.

## 1  Introduction

Methane is a greenhouse gas emitted by a range of natural and anthropogenic sources (Kirschke
et al., 2013; Saunois et al., 2016; Turner et al., 2017). Anthropogenic methane emissions are difficult
to quantify because they tend to originate from a large number of potentially transient point sources
such as livestock operations, oil/gas leaks, landfills, and coal mine ventilation. Atmospheric methane
observations from surface and aircraft have been used to quantify emissions (e.g., Miller et al., 2013;
Caulton et al., 2014; Karion et al., 2013, 2015; Lavoie et al., 2015; Conley et al., 2016; Peischl et al.,
2015, 2016; Houweling et al., 2016) but are limited in spatial and temporal coverage. Satellite
measurements have dense and continuous coverage but limitations from observational errors and
pixel resolution need to be understood. Here we perform an observing system simulation experiment
(OSSE) to investigate the information content of different configurations of satellite instruments for
observing fine-scale and transient methane sources, taking as a test case the oil/gas production sector.
Low-Earth orbit satellite observations of methane by solar backscatter in the shortwave infrared
(SWIR) have been available since 2003 from the SCIAMACHY instrument (2003–2012; Franken-
berg et al., 2005) and from the GOSAT instrument (2009–present; Kuze et al., 2009, 2016). SWIR
instruments measure the atmospheric column of methane with near-unit sensitivity throughout the
troposphere. SCIAMACHY and GOSAT demonstrated the capability for high-precision (<1%)
measurements of methane from space (Buchwitz et al., 2015), but SCIAMACHY had coarse pix-
els ($30 \times 60$ km$^2$ in nadir) and GOSAT has sparse coverage (10-km diameter pixels separated by
250 km). Inverse analyses have used observations from these satellite-based instruments to estimate
methane emissions at ~100-1000 km spatial resolution  (e.g., Bergamaschi et al., 2009, 2013; Fraser
et al., 2013; Monteil et al., 2013; Wecht et al., 2014a; Cressot et al., 2014; Kort et al., 2014; Turner
et al., 2015, 2016a; Alexe et al., 2015; Tan et al., 2016; Buchwitz et al., 2017; Sheng et al., 2017,
2018). But such coarse resolution makes it difficult to resolve individual source types because of
spatial overlap (Maasakkers et al., 2016).
Improved observations of methane from space are expected in the near future (Jacob et al., 2016).
The GHGSat instrument launched in June 2016 (http://www.ghgsat.com/) has $50 \times 50$ m$^2$ effective
pixel resolution over selected $12 \times 12$ km$^2$ viewing scenes with a return time of a few weeks, suitable
for detecting large point sources. The TROPOMI instrument (Veefkind et al., 2012; Butz et al., 2012;
Hu et al., 2016), launched in October 2017, will provide global mapping at $7 \times 7$ km$^2$ nadir resolution
once per day. The GeoCARB geostationary instrument (Polonsky et al., 2014; O'Brien et al., 2016)



will be launched in the early 2020s with current design values of $3\times3$ km$^2$ pixel resolution and
twice-daily return time. Additional instruments are presently in the proposal stage with improved
combinations of pixel resolution, return time, and instrument precision (Fishman et al., 2012; Butz
et al., 2015; Xi et al., 2015).
An OSSE simulates the atmosphere as it would be observed by an instrument with a given ob-
serving configuration and error specification. Several OSSEs have been conducted to evaluate the
potential of satellite observations to quantify methane sources, but they have either been conducted
at coarse ($\sim50\times50$ km$^2$) spatial resolution (Wecht et al., 2014b; Bousserez et al., 2016) or assumed
idealized flow conditions (Bovensmann et al., 2010; Rayner et al., 2014). Jacob et al. (2016) pre-
sented a simple mass balance equation to compare the source detection capabilities of satellite in-
struments with different pixel resolutions, precisions, and return times, but they used information
from the source pixel only and assumed a homogeneous flow. Here we use a 1-week simulation of
atmospheric methane with $1.3\times1.3$ km$^2$ resolution over a $290\times235$ km$^2$ domain to simulate con-
tinuous and transient emissions in the Barnett Shale region of Northeast Texas, and from there we
quantify the capability of different satellite instrument configurations to resolve and quantify these
sources at the kilometer scale.
**2   High-resolution OSSE environment**

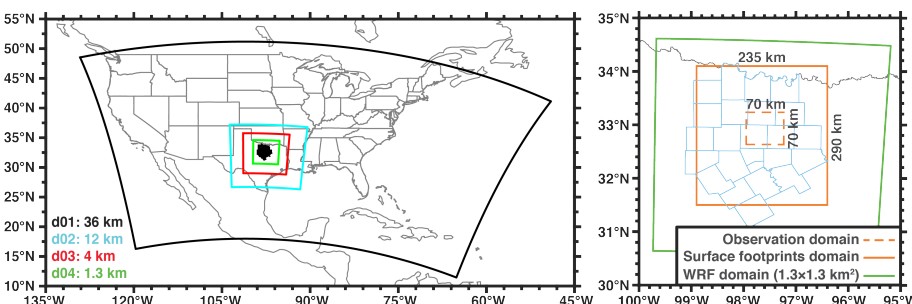

**Fig. 1.** High-resolution OSSE domain. Left panel shows the successive nested WRF domains at 36, 12, 4, and 1.3 km spatial resolutions, with the coarser domains providing initial and boundary conditions for the finer domains. Black shaded region is the Barnett Shale region of Northeast Texas. Right panel shows the domain for the OSSE. Green box is the innermost 1.3 km WRF domain, dashed orange box is the observation domain, solid orange box is the domain over which the footprints are computed. Light blue lines indicate the counties in the Barnett Shale.

We simulate atmospheric methane concentrations over the Barnett Shale of Northeast Texas at
$1.3\times1.3$ km$^2$ horizontal resolution for the period of October 19-25, 2013 using a framework sim-
ilar to that of Turner et al. (2016b). The simulation uses version 3.5 of the Weather Research



and Forecasting (WRF) model (Skamarock et al., 2008) over a succession of nested domains (left
panel in Figure 1) with $1.3\times1.3$ km$^2$ spatial resolution in the innermost domain covering $290\times235$
km$^2$. There are 50 vertical layers up to 100 hPa. Boundary-layer physics are represented with the
Mellor-Yamada-Janic scheme and the land surface is represented with the 5-layer slab model (Ska-
marock et al., 2008). The simulation is initialized with assimilated meteorological observations
from the North American Regional Reanalysis (https://www.ncdc.noaa.gov/data-access/model-data/
model-datasets/north-american-regional-reanalysis-narr). Overlapping 30-hour forecasts were ini-
tialized every 24 hours at 00 UTC and the first 6 hours of each forecast were discarded to allow for
model spinup. Grid nudging was used in the outer-most domain.
WRF meteorology is used to drive the Stochastic Time-Inverted Lagrangian Transport (STILT)
model (Lin et al., 2003). STILT is a Lagrangian particle dispersion model. It advects an ensemble
of particles backward in time from selected receptor locations, using the archived hourly WRF wind
fields and boundary-layer heights. STILT calculates the footprint for the receptors; a spatio-temporal
map of the sensitivity of observations to emissions contributing to the concentration at each selected
receptor location and time. We use STILT to calculate 10-day footprints for hourly column concen-
trations at $1.3\times1.3$ km$^2$ resolution over a $70\times70$ km$^2$ domain in the innermost WRF nest, tracking
the resulting footprints over a $290\times235$ km$^2$ domain (right panel in Figure 1). With this system we
examine the constraints on emissions over the $290\times235$ km$^2$ domain provided by dense SWIR satel-
lite observations (over the $70\times70$ km$^2$ domain) that have up to 1.3 km pixel resolution and hourly
daytime frequency. Footprints for each column are obtained by releasing 100 STILT particles from
vertical levels centered at 28 m above the surface, 97 m, 190 m, 300 m, and 8 additional levels up
to 14 km altitude spaced evenly on a pressure grid. The column footprints are then constructed by
summing the pressure-weighted contributions from individual levels, using a typical SWIR averag-
ing kernel taken from Worden et al. (2015) with near-uniformity in the troposphere, and correcting
for water vapor (see Appendix A in O'Dell et al., 2012).
The footprint for the $i^{\text{th}}$ receptor location and time can be expressed as a vector $\mathbf{h}_i = (\partial y_i/\partial \mathbf{x})^T$
describing the sensitivity of the column concentration $y$ at that receptor location and time to the
emission fluxes $\mathbf{x}$ over the $290\times235$ km$^2$ domain and previous times extending up to 10 days. Here
$\mathbf{x}$ is arranged as a vector of length $n$ assembling all the emission grid cells and hours, allowing the
emissions to vary on an hourly basis. The column concentration is expressed as the dry air column-
average mixing ratio (ppb) following common practice (Jacob et al., 2016). The emissions $\mathbf{x}$ have
units of nmol m$^{-2}$ s$^{-1}$, so that the footprint has units of ppb nmol$^{-1}$ m$^2$ s. The column concentration
for the $i^{\text{th}}$ observation ($y_i$) can be reconstructed from its footprint as:

$$y_i = \mathbf{h}_i \mathbf{x} + b_i \qquad (1)$$

where $b_i$ is the background column concentration upwind of the $290\times235$ km$^2$ domain. We can
then write the full set of observations as a vector $\mathbf{y}$ of length $m$, and reshape the set of $m$ footprint





vectors $\mathbf{h}$ into an $m \times n$ sparse matrix $\mathbf{H} = \partial \mathbf{y}/\partial \mathbf{x}$ (where $m$ is the number of observations and $n$ is
the number of state vector elements):

$$\mathbf{y} = \mathbf{H}\mathbf{x} + \mathbf{b} \qquad (2)$$

where $\mathbf{b}$ is the background vector with elements $b_i$ and $\mathbf{H}$ is the Jacobian matrix that maps emissions
to concentration enhancements due to emissions within our domain.

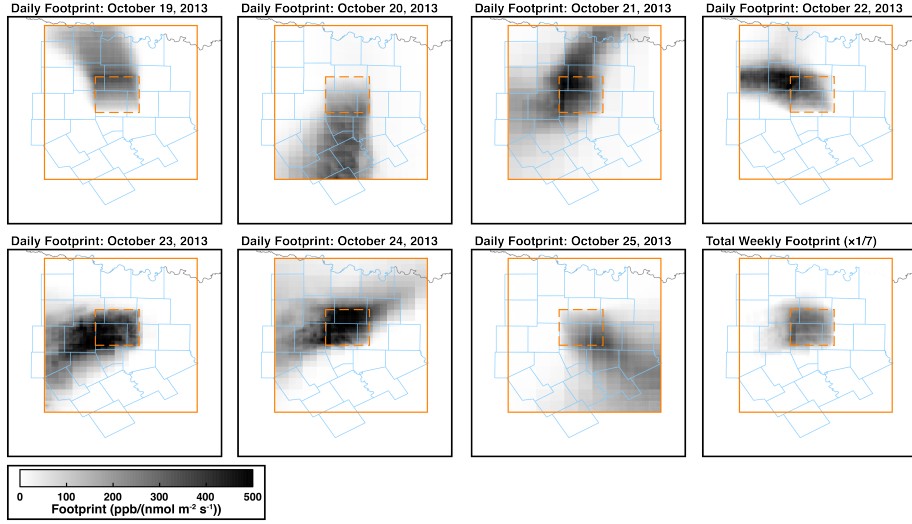

**Fig. 2.** Summed methane column footprints for all $1.3 \times 1.3$ km$^2$ grid cells in the $70 \times 70$ km$^2$ observation domain defined by the dashed orange box. The footprints are calculated from 8 to 17 local time over the $290 \times 235$ km$^2$ domain defined by the solid orange box. Bottom right panel shows the summed footprint for the full week, scaled by $1/7$.

Figure 2 shows the sum of all column footprints produced on individual days for the $70 \times 70$ km$^2$
observation domain. The footprints show large variability from day to day over the course of the
week, reflecting meteorological variability. For example, winds are from the north on October 19th
and from the south on October 20th. The winds are weak on October 24th, resulting in a strong
local contribution to the footprint. Summing the footprints over the course of the week (bottom right
panel of Fig. 2), we find that the observations are strongly sensitive to the core $70 \times 70$ km$^2$ domain
with a diffuse sensitivity over the outer $290 \times 235$ km$^2$ domain.
The footprint information can be combined with an emission inventory for the $290 \times 235$ km$^2$
domain to generate a field of column concentrations over the $70 \times 70$ km$^2$ domain as would be ob-
served from satellite. We use for this purpose the Environmental Defense Fund (EDF) inventory for
the Barnett Shale in October 2013 at $4 \times 4$ km$^2$ resolution compiled by Lyon et al. (2015). We down-
scale the EDF inventory by uniform attribution from $4 \times 4$ km$^2$ to $1.3 \times 1.3$ km$^2$ spatial resolution.




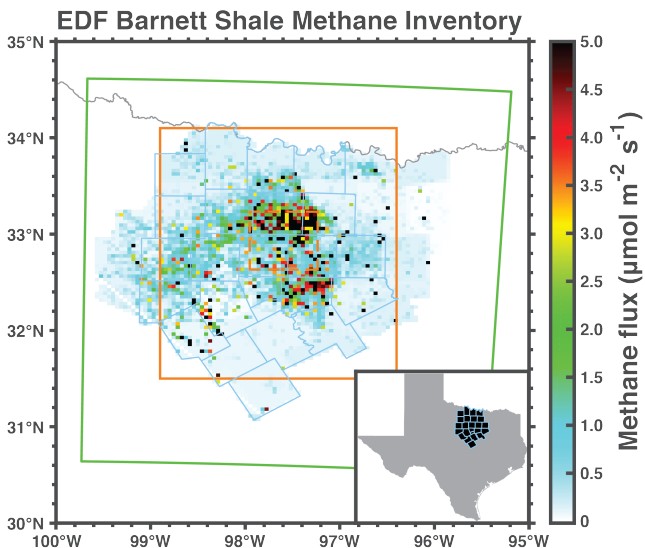

**Fig. 3.** Gridded Environmental Defense Fund (EDF) methane emission inventory for the Barnett Shale in Northeast Texas in October 2013 (Lyon et al., 2015). Spatial resolution is $4\times4$ km$^2$. White areas are outside the inventory domain.

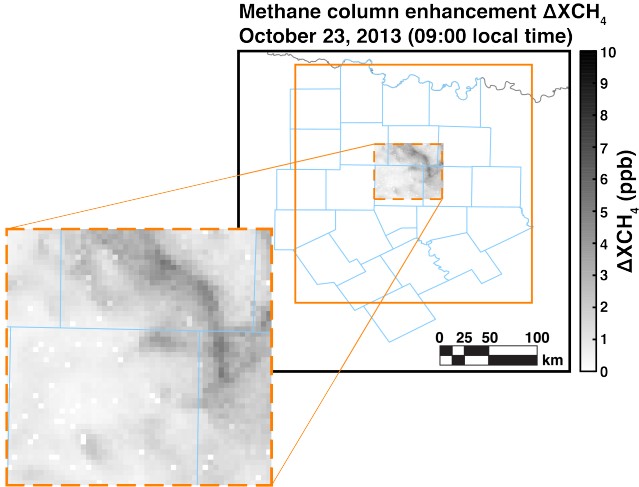

**Fig. 4.** Simulated methane concentration enhancements relative to background ($\Delta XCH_4 = \mathbf{Hx}$) in the $70\times70$ km$^2$ observation domain of the Barnett Shale (dashed orange box), as derived from the downscaled EDF methane inventory ($\mathbf{x}$) and the WRF-STILT footprints ($\mathbf{H}$) within the $290\times235$ km$^2$ OSSE domain (solid orange box). Values are for October 23 at 9 local time.





**Table 1.** Satellite observing systems considered in this work.

| Instrument | Observation Frequency[a] | Pixel resolution (km$^2$) | Precision (ppb) |
|---|---|---|---|
| hi-res[b] | hourly | $1.3 \times 1.3$ | 1.0 |
| GeoCARB (hourly) | hourly | $2.7 \times 3.0$ | 4.0 |
| GeoCARB | twice daily | $2.7 \times 3.0$ | 4.0 |
| GeoCARB (daily) | daily | $2.7 \times 3.0$ | 4.0 |
| TROPOMI | daily | $7.0 \times 7.0$ | 10.8 |

[a] Hourly observations are 10 times per day at 8-17 local time, twice daily observations are at 10 and 14 local time, and daily observations are at 13 local time.

[b] Aspirational instrument with the highest observation frequency and pixel resolution that can be simulated within our OSSE framework.

The inventory is shown in Fig. 3 and includes contributions from oil/gas production, livestock op-
erations, landfills, and urban emissions from the Dallas-Fort Worth area. It provides mean monthly
values with no temporal resolution, but presumes that some sources will behave as sporadic large
transients (Zavala-Araiza et al., 2015). Figure 4 shows an example of the methane column enhance-
ments above background (**Hx**) computed at 9 local time on October 23. We find enhancements
in the range of 0-10 ppb due to emissions within the $290 \times 235$ km$^2$ OSSE footprint domain. In
what follows we will examine the potential of different satellite observing systems to detect these
enhancements relative to the background and interpret them in terms of local sources.

## 3  Information content of different satellite observing systems

We aim to determine the information content from different satellite-based observing systems regard-
ing the spatial and temporal distribution of emissions in the Barnett Shale. We consider both steady
and potentially transient emissions with 5 different satellite observing configurations (Table 1).
TROPOMI (global daily mapping, $7 \times 7$ km$^2$ nadir pixel resolution, 11 ppb precision; Veefkind et al.,
2012) was launched in October 2017 and is expected to provide an operational data stream by the end
of 2018. GeoCARB (geostationary, $2.7 \times 3.0$ km$^2$ pixel resolution, 4 ppb precision; O'Brien et al.,
2016) is planned for launch in the early 2020s and its observation schedule is still under discussion
with a tentative design for observations twice daily; here we examine different return frequencies of
hourly, twice daily, and daily. Finally, the hypothetical "hi-res" configuration assumes geostation-
ary hourly observations at the $1.3 \times 1.3$ km$^2$ pixel resolution of our WRF simulation and with 1 ppb
precision; it represents an aspirational system that combines the frequent return time, fine pixel res-
olution, and high precision of instruments presently at the proposal stage (Bovensmann et al., 2010;
Fishman et al., 2012; Xi et al., 2015). All configurations are filtered for cloudy scenes.





The various satellite observing configurations of Table 1 differ in their return frequency, pixel
resolution, and instrument precision. The benefit of improving any of these attributes may be lim-
ited by error in the forward model used in the inverse analysis (i.e., the Jacobian matrix $\mathbf{H}$) and by
spatial or temporal correlation of the errors. These limitations are described by the model-data mis-
match error covariance matrix ($\mathbf{R}$) including summed contributions from the instrument, forward
model, and representation errors (Turner and Jacob, 2015; Brasseur and Jacob, 2017). Represen-
tation errors are negligible here because the instrument pixels are commensurate or coarser than
the model grid resolution. Instrument error (i.e., precision) is listed in Table 1. Forward model
error is estimated by computing STILT footprints for a subset of the meteorological period using
the Global Data Assimilation System (GDAS; https://www.ncdc.noaa.gov/data-access/model-data/
model-datasets/global-data-assimilation-system-gdas), applying the two sets of footprints to either
the EDF methane inventory (Fig. 3; Lyon et al., 2015) or the gridded EPA inventory (Maasakkers
et al., 2016), and computing semivariograms of differences in column concentrations. From this we
obtain a forward model error standard deviation of 4 ppb with an error correlation length scale of 40
km. We assume a temporal model error correlation length of 2 hours. Sheng et al. (2017) previously
derived a temporal model error correlation length of 5 hours in simulation of TCCON methane col-
umn observations at 25 km resolution, and we expect our correlation length to be shorter because of
the finer resolution.
Bayesian inference is commonly used when estimating methane emissions with atmospheric ob-
servations, allowing for errors in the observations and in the prior estimates:

$$P(\mathbf{x}|\mathbf{y}) \propto P(\mathbf{y}|\mathbf{x})P(\mathbf{x}) \qquad (3)$$

where $P(\mathbf{x}|\mathbf{y})$ is the posterior probability density function (pdf) of the state vector ($\mathbf{x}$) given the
observations ($\mathbf{y}$), $P(\mathbf{y}|\mathbf{x})$ is the conditional pdf of $\mathbf{y}$ given $\mathbf{x}$, and $P(\mathbf{x})$ is the prior pdf of $\mathbf{x}$. A
common assumption is that $P(\mathbf{y}|\mathbf{x})$ and $P(\mathbf{x})$ are normally distributed which allows us to write the
posterior pdf as

$$P(\mathbf{x}|\mathbf{y}) \propto \exp\left\{ -\frac{1}{2}(\mathbf{y} - \mathbf{H}\mathbf{x})^T \mathbf{R}^{-1}(\mathbf{y} - \mathbf{H}\mathbf{x}) - \frac{1}{2}(\mathbf{x} - \mathbf{x}_a)^T \mathbf{B}^{-1}(\mathbf{x} - \mathbf{x}_a) \right\} \qquad (4)$$

where $\mathbf{B}$ is the $n \times n$ prior error covariance matrix and $\mathbf{x}_a$ is the $n \times 1$ vector of prior fluxes. The
most probable solution is obtained by minimizing the cost function:

$$\mathcal{J}(\mathbf{x}) = \frac{1}{2}(\mathbf{y} - \mathbf{H}\mathbf{x})^T \mathbf{R}^{-1}(\mathbf{y} - \mathbf{H}\mathbf{x}) + \frac{1}{2}(\mathbf{x} - \mathbf{x}_a)^T \mathbf{B}^{-1}(\mathbf{x} - \mathbf{x}_a) \qquad (5)$$

yielding the posterior estimate ($\hat{\mathbf{x}}$):

$$\hat{\mathbf{x}} = \mathbf{x}_a + \underbrace{\left(\mathbf{H}^T \mathbf{R}^{-1}\mathbf{H} + \mathbf{B}^{-1}\right)^{-1}}_{\text{posterior covariance matrix}} \mathbf{H}^T \mathbf{R}^{-1}(\mathbf{y} - \mathbf{H}\mathbf{x}) \qquad (6)$$



**Table 2.** Cost functions for different formulations of the inverse problem[a].

| Method | Cost function |
|---|---|
| Least-squares regression | $(\mathbf{y}-\mathbf{Hx})^T\mathbf{R}^{-1}(\mathbf{y}-\mathbf{Hx})$ |
| LASSO regression | $(\mathbf{y}-\mathbf{Hx})^T\mathbf{R}^{-1}(\mathbf{y}-\mathbf{Hx})+\gamma\sum_i|x_i|$ |
| Tikhonov regularization | $(\mathbf{y}-\mathbf{Hx})^T\mathbf{R}^{-1}(\mathbf{y}-\mathbf{Hx})+\gamma\mathbf{x}^T\mathbf{x}$ |
| Bayesian inference, Gaussian | $(\mathbf{y}-\mathbf{Hx})^T\mathbf{R}^{-1}(\mathbf{y}-\mathbf{Hx})+(\mathbf{x}-\mathbf{x}_a)^T\mathbf{B}^{-1}(\mathbf{x}-\mathbf{x}_a)$ |
| Geostatistical inverse model | $(\mathbf{y}-\mathbf{Hx})^T\mathbf{R}^{-1}(\mathbf{y}-\mathbf{Hx})+(\mathbf{x}-\mathbf{G\beta})^T\mathbf{B}^{-1}(\mathbf{x}-\mathbf{G\beta})$ |

[a]$\gamma$ is the regularization parameter for LASSO regression and Tikhonov regularization. $\mathbf{G}$ is a matrix with columns corresponding to different spatial datasets and $\boldsymbol{\beta}$ is a vector of drift coefficients for the spatial datasets. Other variables defined in the text.

with an $n \times n$ posterior error covariance matrix:

$$\mathbf{Q} = (\underbrace{\mathbf{H}^T\mathbf{R}^{-1}\mathbf{H}}_{\text{observations}} + \underbrace{\mathbf{B}^{-1}}_{\text{prior}})^{-1} \tag{7}$$

that characterizes the uncertainty in the solution. The first term in the posterior covariance ma-
trix is known as the Fisher information matrix: $\mathcal{F} = \mathbf{H}^T\mathbf{R}^{-1}\mathbf{H}$ (see, for example, Rodgers, 2000;
Tarantola, 2004). $\mathcal{F}$ defines the observational contribution to the posterior uncertainty.
Comparison between $\mathcal{F}$ and $\mathbf{B}^{-1}$ identifies the extent to which the observations reduce the un-
certainty in the fluxes. Specifically, the number of pieces of information on emissions acquired to
better than measurement error is the number of eigenvalues of $\mathbf{B}^{1/2}\mathcal{F}\mathbf{B}^{1/2}$ that are greater than
unity (Rodgers, 2000). As such, the Fisher information matrix and prior error covariance matrix can
quantify the effective rank of the observing system.
A drawback with this formulation of the information content is that it relies on the assumption of
a Gaussian prior pdf. A number of papers have suggested that the pdf of methane emissions from a
given source may be skewed, with a "fat tail" of transient high emissions (e.g., Brandt et al., 2014;
Zavala-Araiza et al., 2015; Frankenberg et al., 2016). Alternate formulations for the cost function
to be minimized may include no prior information (least-squares regression), a prior constraint that
promotes a sparse solution (e.g., Candes and Wakin, 2008), a prior constraint based on frequen-
tist regularization approaches (such as LASSO regression or Tikhonov regularization), or a prior
constraint based on the spatial patterns of emissions rather than their magnitudes (geostatistical in-
version). Table 2 lists the corresponding formulations. From Table 2 we see that the observation term
is the same in all cases. Thus the Fisher information matrix provides a general measure of the in-
formation content provided by an observing system, independent of the form of the prior constraint,
and we use it in what follows as a measure of the information content.
The Fisher information matrix is an $n \times n$ matrix. Each of its $n$ eigenvectors represent an inde-
pendent normalized emission flux pattern and the corresponding eigenvalues are the inverses of the
error variances associated with that pattern. A more useful way of stating this is that the inverse





square root of the $i^{\text{th}}$ eigenvalue of $\mathcal{F}$ represents the flux threshold $f_i$ needed for the observations
to be able to constrain the emission flux pattern represented by the $i^{\text{th}}$ eigenvector. Whether that
flux threshold is useful depends on the magnitude of the emissions, and this can be assessed for the
problem at hand. Thus the eigenanalysis of the Fisher information matrix gives us a general estimate
of the capability of an observing system to quantify emissions, which can then be applied to any
actual $n \times n$ emission field.

For a given emission field, we may expect that some of the $n$ emission flux patterns will be

usefully constrained by the observing system while others are not. The number of patterns that are
usefully constrained represents the number $\mathcal{I} \leq n$ pieces of information on emissions provided by
the observing system. We will equivalently refer to it as the rank of the Fisher information matrix.
This is determined by comparing the eigenvalues of an emission inventory ($e_i$) to the flux thresholds.
The number of $e_i$ larger than the corresponding $f_i$ provides a cut-off to estimate $\mathcal{I}$:

$$\mathcal{I} = \sum_i \begin{cases} 1, & e_i > f_i \\ 0, & e_i \leq f_i \end{cases} \tag{8}$$

In the case of Bayesian inference, this is roughly equivalent to the degrees of freedom for signal with
a diagonal prior error covariance matrix and a relative uncertainty of 100%. But the eigenanalysis
of the Fisher information matrix provides a more general approach of the capability of an observ-
ing system that can be confronted to any prior constraint and allows intercomparison of different
observing system configurations.

There is an inconsistency in this formulation of $\mathcal{I}$: $\mathcal{F}$ and $\mathbf{B}^{-1}$ have different eigenspaces. In this

work we have chosen to treat these matrices separately because, in practice, it is computationally
infeasible to directly compute the eigenvalues of the matrix product if $n$ is large, as in the case here
of constraining hourly emissions of the spatially distributed inventory. This inconsistency results in
our estimate of $\mathcal{I}$ likely being an upper bound on the information content (see Appendix for details).

## 4   Comparing different satellite configurations

The eigenanalysis of Section 3 allows us to intercompare the value of different satellite configura-
tions for resolving the fine-scale patterns of methane emissions within a given domain. Here we
apply it to the Barnett Shale domain of Section 2. We consider two limiting cases: Case #1 assumes
the emissions to be temporally invariant and Case #2 assumes the emissions to vary hourly with no
temporal correlation. In Case #1 the problem is typically overdetermined ($m > n$), depending on
the satellite configuration, and the maximum rank of $\mathcal{F}$ is $n$ (the number of emission grid cells). In
Case #2 the problem is underdetermined ($m < n$) and the maximum rank of $\mathcal{F}$ is $m$ (the number of
observations).

In both Case #1 and #2, the observations only provide useful information (as defined by Eq. 8) if

the signal is larger than the noise, as diagnosed by the $e_i > f_i$ criterion of Eq. 8. Here the emissions


are the downscaled EDF inventory, which includes 40,140 grid cells in the $290 \times 235$ km$^2$ inversion
domain ($n = 40,140$ in Case #1 with temporally invariant emissions) but only 2,601 of those grid
cells are within the $70 \times 70$ km$^2$ observation domain (dashed orange box in Fig. 1) where we might
expect the observations to provide the strongest constraints. In Case #2 with temporally variable
emissions we have $n = 40,140 \times 24 = 963,360$ grid cells for a single day.

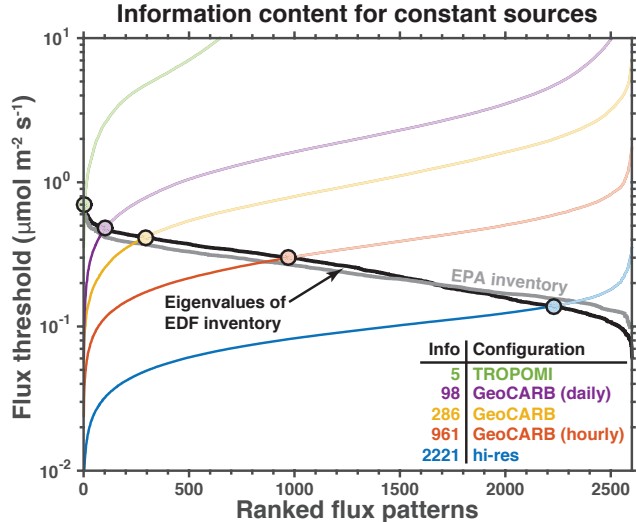

**Fig. 5.** Capability of different configurations for satellite observations of atmospheric methane (Table 1) to resolve the fine-scale ($1.3 \times 1.3$ km$^2$) patterns of variability of temporally invariant emissions in a $290 \times 235$ km$^2$ domain and for a 1-week observation period. The colored lines show the flux thresholds for the different emission patterns of variability in the domain, as given by the ordered inverse square roots of the eigenvalues of the Fisher information matrix. Solid black line is the eigenvalues of the emissions from the EDF Barnett Shale methane inventory (Lyon et al., 2015) and the solid gray line is the gridded EPA inventory. The region above the black line is where the noise is larger than the signal. Filled circles indicate the information content of the observing system ($\mathcal{I}$) for a given satellite configuration at $1.3 \times 1.3$ km$^2$ spatial resolution. Inset table lists the information contents for the five configurations.

Figure 5 shows the ensemble of flux thresholds for the five satellite configurations, assuming
temporally invariant emissions. The ranked flux patterns are on the abscissa; leading flux patterns
correspond to larger patterns of variability (e.g., regional-scale emissions), and the trailing flux pat-
terns correspond to fine-scale variability. The corresponding flux thresholds are on the ordinate.
The flux threshold is lowest for the leading flux patterns and largest for the trailing flux patterns.
This means that the regional-scale emissions are easiest to quantify and the finer-scale emissions are
increasingly difficult to quantify. The information content ($\mathcal{I}$) is obtained from the intersection of
the flux thresholds (colored lines) with the eigenvalues from the emission inventory (black line). A





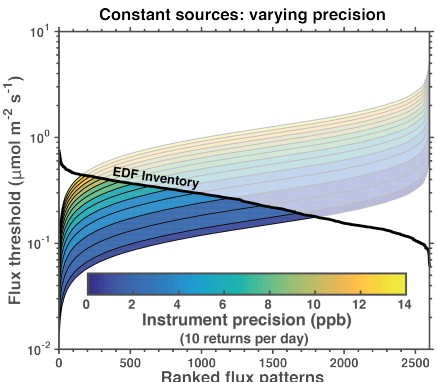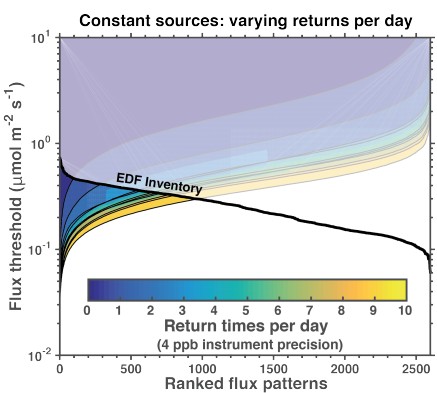

**Fig. 6.** Capability of GeoCARB-like satellite configurations to resolve the fine-scale ($1.3 \times 1.3$ km$^2$) patterns of variability of temporally invariant emissions in a $290 \times 235$ km$^2$ domain and for a 1-week observation period. Left panel shows the results for a configuration with 10 returns per day (hourly observations) where the instrument precision is varied from 0 to 14 ppb. Right panel shows the results for a configuration with 4 ppb instrument precision and the return frequency per day is varied from 1 to 10. Solid black line shows eigenvalues of the EDF Barnett Shale methane emission inventory (Lyon et al., 2015). The region above the black line is where the noise is larger than the signal.

higher information content means that finer scales of emission variability can be detected.
From Fig. 5, we see that a week of TROPOMI observations provides 5 pieces of information,
indicating that TROPOMI should be able to constrain the mean emissions from the Barnett Shale
and the coarse spatial distribution of these emissions. The three GeoCARB configurations provide
98–961 pieces of information dependent on whether the observations are daily, twice daily, or hourly.
Hourly observations provide 10 times more information (as defined by Eq. 8) on emission patterns
than daily observations, and 3 times more than twice-daily observations (the default configuration
of GeoCARB). Remarkably, more is gained by going from daily to twice-daily (factor of 3.4) than
going from twice-daily to hourly (factor of 2.9), because of the temporal error correlation in the
transport model. The aspirational hi-res satellite configuration provides 2,221 pieces of information
on temporally invariant sources, corresponding to 85% of the flux patterns, which means that much
of the spatial variability in the $1.3 \times 1.3$ km$^2$ emissions in the Barnett Shale is resolved.
Figure 6 further quantifies the importance of instrument precision and return frequency for the
GeoCARB pixel resolution of $2.7 \times 3.0$ km$^2$. It shows the flux thresholds for a set of configurations
where the instrument precision is varied from 0 to 14 ppb and the return frequency is varied from 1
to 10 returns per day. We find that instrument precision is more important than return frequency for
increasing the information content from the observations.
In Case #2 we assume that the methane sources in individual pixels vary in time on an hourly
basis with no correlation from one hour to the next, making the problem generally underdetermined





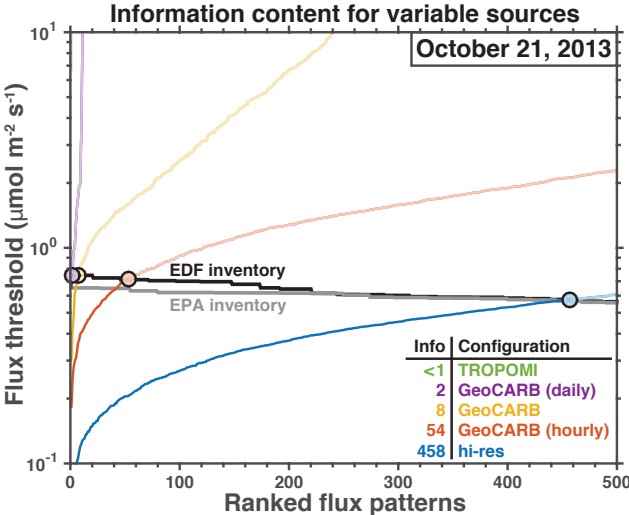

**Fig. 7.** Same as Fig. 5 but for temporally variable sources on October 21, 2013.

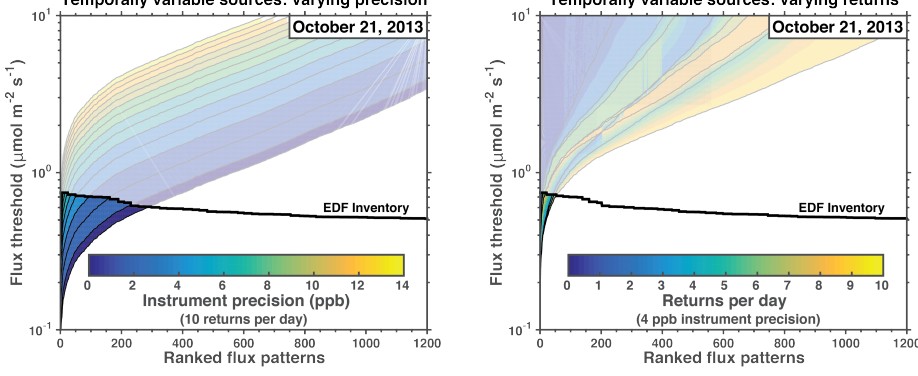

**Fig. 8.** Same as Fig. 6 but for temporally variable sources on October 21, 2013.





($m < n$) for all satellite configurations. Here we aim to determine the ability of the satellite obser-
vations to quantify the hourly emissions over the spatial patterns defined by the eigenvectors of $\mathcal{F}$
and making no assumption as to the persistence of those emissions. We treat each day independently
and compute the eigenvalues of the Fisher information matrix for each day. Figure 7 shows the
flux thresholds for the five satellite configurations on a representative day. From Fig. 7, we see that
TROPOMI is unable to provide any information on hourly emissions in the Barnett Shale. The three
GeoCARB configurations provide 2–54 pieces of information. Fig. 8 evaluates the impact of sam-
pling frequency and instrument precision for the GeoCARB configurations. As with the temporally
invariant case, we find that instrument precision is more important for increasing the information
content. The aspirational "hi-res" configuration (shown in Fig. 7) is the only configuration that is
able to provide substantial information (458 pieces of information) on temporally variable emissions.
Figure 9 summarizes the findings from Figs. 6 and 8. It compares the information content $\mathcal{I}$ from
configurations with $2.7 \times 3.0$ km$^2$ spatial resolution (GeoCARB) as the instrument precision and
return frequency are varied from 0 to 14 ppb and 1 to 10 returns per day, respectively, for both tem-
porally variable and constant sources. Uncertainty on $\mathcal{I}$ is estimated by randomly sampling $e_i$ from
the ensemble of emission inventory eigenvalues and comparing to $f_i$ in Eq. 8. For the temporally
invariant sources (Case #1), we find considerable increases in information content for instrument pre-
cisions better than 6 ppb (top left panel in Fig. 9) and an approximately linear relationship between
information content and return frequency (top right panel in Fig. 9). The satellite configurations
provide considerably less information for the temporally variable sources (Case #2). We find that
satellite configurations with an instrument precisions worse than 6 ppb provide no information on
temporally variable sources (bottom left panel in Fig. 9). As with the temporally invariant case, we
find an approximately linear relationship between information content and return frequency (bottom
right panel in Fig. 9). From this, we conclude that a GeoCARB-like instrument would greatly benefit
from having an instrument precision better than 6 ppb.
## 5   Conclusions
We conducted an observing system simulation experiment (OSSE) to evaluate the potential of dif-
ferent satellite observation systems for atmospheric methane to quantify methane emissions at kilo-
meter scale. This involved a 1-week WRF-STILT simulation of atmospheric methane columns with
$1.3 \times 1.3$ km$^2$ spatial resolution over a $290 \times 235$ km$^2$ domain (Barnett Shale of Northeast Texas)
to quantify the information content of different satellite instrument configurations for resolving the
kilometer-scale distribution of methane emissions within that domain. We evaluated the information
content of the different satellite observing systems through an eigenanalysis of the Fisher informa-
tion matrix $\mathcal{F}$, which characterizes the capability of an observing system independently of the form
of the prior information. The eigenvalues of $\mathcal{F}$ define the emission flux thresholds for detection of





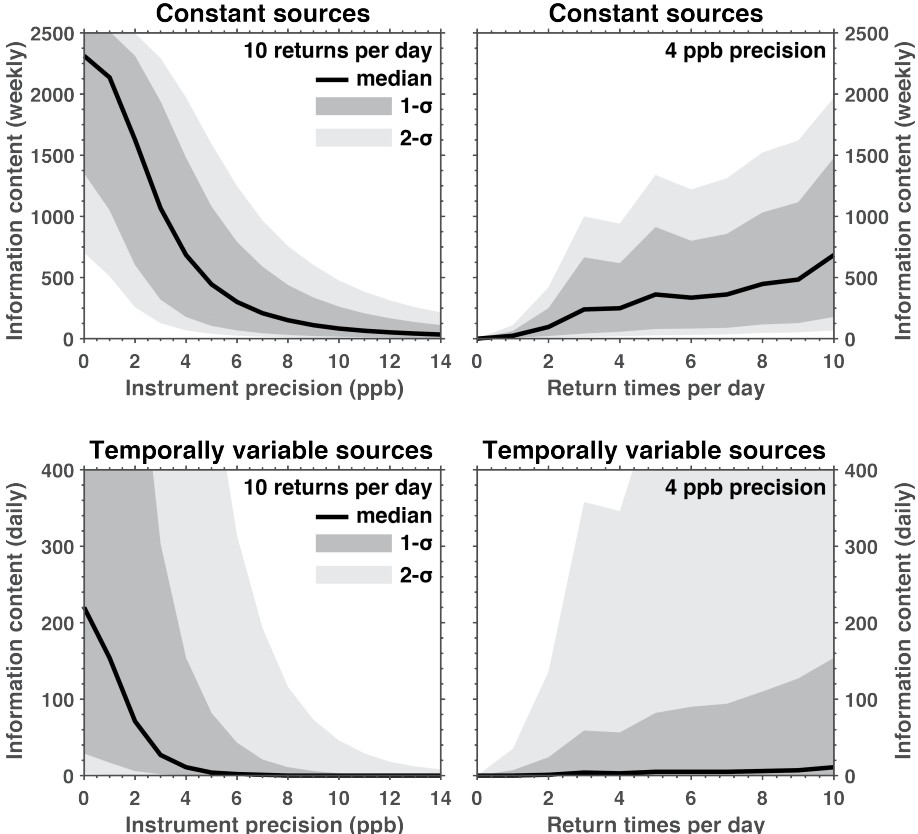

**Fig. 9.** Information content $\mathcal{I}$ as a function of the instrument precision (left column) and the sampling frequency per day (right column) for a satellite with a pixel resolution of $2.7 \times 3.0 \text{ km}^2$. Top row is for Case #1 where the sources are assumed to be temporally invariant and bottom row is for Case #2 where the sources are temporally variable. Solid black line is the median information content. A 4 ppb model error is included, see Section 3. Uncertainty is from randomly sampling $e_i$ from the eigenvalues of the EDF inventory.





emission patterns down to 1.3 km in scale as defined by the eigenvectors. Here we put these flux
thresholds in context of the high-resolution EDF emission inventory for the Barnett Shale to quantify
the information content from different satellite observing configurations. The same approach could
be readily used for different observation domains and different prior inventories.
We find from this analysis that the recently launched TROPOMI satellite instrument (low Earth
orbit, 7×7 km$^2$ pixels, daily return time, 11 ppb precision) should be able to constrain the mean
emissions in the Barnett Shale and provide some coarse-resolution information on the distribution
of emissions. The planned GeoCARB instrument (geostationary orbit, 2.7×3.0 km$^2$ pixels, twice-
daily return time, 4 ppb precision), will provide 50 times more information than TROPOMI. The
observing frequency of GeoCARB is still under discussion; we find that twice-daily observations
triple the information content relative to daily observations, while hourly observations allow another
tripling. The 4 ppb precision of GeoCARB is well adapted to the magnitude of methane sources;
we find that a precision larger than 6 ppb would considerably decrease the information content. An
aspirational "hi-res" instrument using attributes of currently proposed instruments (geostationary
orbit, 1.3×1.3 km$^2$ pixels, hourly return time, 1 ppb precision) can resolve much of the kilometer-
scale spatial distribution in the EDF inventory. This assumes that the emissions are constant in time
or that their temporal variability is known. Resolving hourly variable emissions at the kilometer-
scale will be very limited even with the aspirational "hi-res" instrument.
**Appendix Computing the information content**
We treat $\mathcal{F}$ and $\mathbf{B}^{-1}$ separately because it is computationally infeasible to compute the eigenval-
ues of the matrix product when we attempt to resolve hourly emissions as $n > 10^6$ and both $\mathcal{F}$
and $\mathbf{B}^{-1}$ are $n \times n$ matrices. This separation of $\mathcal{F}$ and $\mathbf{B}^{-1}$ results in our estimate of $\mathcal{I}$ likely be-
ing an upper bound on the information content. This follows from Bhatia (1997) who prove that
$\lambda(\mathbf{CD}) \prec_w \lambda^\downarrow(\mathbf{C}) \cdot \lambda^\downarrow(\mathbf{D})$, where $\mathbf{C}$ and $\mathbf{D}$ are Hermitian positive definite matrices, $\lambda^\downarrow(\mathbf{X})$ de-
notes the vector of eigenvalues of $\mathbf{X}$ in decreasing order, $\prec_w$ is the weak majorization preorder,
and $\mathbf{p} \cdot \mathbf{q} = (p_1 q_1, \ldots, p_n, q_n)$. Therefore, directly computing the eigenvalues of $\mathbf{B}^{1/2} \mathcal{F} \mathbf{B}^{1/2}$, as
Rodgers (2000) suggests for the Bayesian inference case with Gaussian errors, would likely yield
fewer eigenvalues larger than unity than our estimate.
In the case of temporally variable emissions, the system is generally underdetermined ($m < n$)
and we can use a singular value decomposition to efficiently compute the eigenvalues of $\mathcal{F}$. For an
$m \times n$ real matrix $\mathbf{A}$, the non-zero singular values of $\mathbf{A}^T \mathbf{A}$ and $\mathbf{A}\mathbf{A}^T$ are identical (see, for example,
Rodgers, 2000) but the dimensions of these two matrices are $n \times n$ and $m \times m$, respectively, and the
eigenvalues can be computed from the square root of the non-zero singular values. We can write
$\mathcal{F} = \hat{\mathbf{H}}^T \hat{\mathbf{H}}$ where $\hat{\mathbf{H}} = \mathbf{L}^{-1}\mathbf{H}$ is the pre-whitened Jacobian and $\mathbf{L}$ is a lower triangular matrix from
a Cholesky decomposition of $\mathbf{R}$ (such that $\mathbf{R} = \mathbf{L}\mathbf{L}^T$). Thus, the eigenvalues of $\mathcal{F}$ can be obtained



by analysis of either $\hat{\mathbf{H}}^T\hat{\mathbf{H}}$ (an $n \times n$ matrix) or $\hat{\mathbf{H}}\hat{\mathbf{H}}^T$ (an $m \times m$ matrix). Analysis of $\hat{\mathbf{H}}\hat{\mathbf{H}}^T$ does
not yield the eigenvectors of $\mathcal{F}$.
*Acknowledgements.* This work was supported by the ExxonMobil Research and Engineering Company and
the US Department of Energy (DOE) Advanced Research Projects Agency – Energy (ARPA-E). A. J. Turner
is supported as a Miller Fellow with the Miller Institute for Basic Research in Science at UC Berkeley. This
research used the Savio computational cluster resource provided by the Berkeley Research Computing program
at the University of California, Berkeley (supported by the UC Berkeley Chancellor, Vice Chancellor for Re-
search, and Chief Information Officer). This research also used resources from the National Energy Research
Scientific Computing Center, which is supported by the Office of Science of the U.S. Department of Energy
under Contract No. DE-AC02-05CH11231. We also acknowledge high-performance computing support from
Cheyenne (doi:10.5065/D6RX99HX) provided by NCAR's Computational and Information Systems Labora-
tory, sponsored by the National Science Foundation.



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
