# Peer review of "Assessing the capability of different satellite observing configurations to resolve the distribution of methane emissions at kilometer scales"

_Atmospheric Chemistry and Physics, 2018_

## Referee Comment (RC1) · Anonymous Referee #1 · 14 Mar 2018

This paper describes results from an observing system simulation experiment (OSSE) to assess the potential to quantify sources of methane from different satellite observing systems. The focus is on emissions at the kilometre scale, both constant in time and transient. The paper is well written, and I recommend publishing after the following minor comments are addressed.

General comments:

It is amazing to see the enormous improvement in capabilities over the past decades (and in the upcoming decade) to quantify emissions from space with ever increasing spatial (and temporal) resolution. However, one might ask, where the limit would be

e.g. in spatial resolution to retrieve useful information e.g. in the context of mitigation. Given the focus on the kilometre scale, and on temporal variations down to hourly, why not go to even smaller scales? I suggest this should be discussed in the introduction to better motivate the targeted spatial and temporal scale.

Specific comments:

Fig. 4: The methane enhancement looks somewhat patchy, with a number of white pixels with zero or near-zero enhancement next to pixels with significantly larger enhancements. Given that the atmosphere due to advection and mass conservation is expected to be continuous in those enhancements, and given that the grey-scale used for the visualization does not have any step changes, the figure is surprising.

Fig. 6, right panel: the colour regions don't follow the lines as they should (and as they do in the left panel).

L334-336: Please clarify: you state "Analysis of $H^H^T$ does not yield the eigenvectors of F", but the previous sentence states otherwise.

---

## Referee Comment (RC2) · Anonymous Referee #2 · 22 Mar 2018

Summary/General comments: Turner et al. present an OSSE to assess the performance of different space-based methane measurements (TROPOMI, GeoCarb, aspirational), in particular considering the ability of these different sensors to evaluate methane emissions from the Barnett Shale, a major oil and gas production region in US. This manuscript is very well written – clear, concise, and presents interesting results of particular relevance at this junction in time. I'm supportive of publication once my minor concerns mentioned below are addressed.

Larger context issue: The work as presented lacks some context that limits the extent and value of the conclusions. This could be addressed easily and would make the

[Figure]

work far more impactful. What I would like to see is more quantitative and qualitative assessment of what the Barnett shale region looks like as a source region compared to other regions and sources of methane. Is the Barnett a typical oil/gas field (for the US, for the globe)? Are emissions particularly large (or small) from this region? Are emissions particularly spatially heterogeneous (lot of intense point sources? Heavy-tail distribution of emissions?)? How does this compare to other interesting methane source regions? Would results be extensible to other oil/gas regions? To regions with intense wetlands? The work presented in convincing for the capabilities/limitations of different sensors—but I don't know if the 6ppb suggested observational threshold is actually an important threshold for studying any domain other than the Barnett.

Question on methodology: What is the impact of choosing to only simulate observations made within the region defined (dashed orange box in Fig. 2)? All the sensors considered would make observation surrounding this box as well, which would have overlapping sensitivity with this region. How does neglecting these observations impact the results? In particular, for sensors like TROPOMI with 'coarser' resolution, might the use of these observation points actually improve the results?

Minor comments (predominantly asking for more specifics/clarifications in abstract): Line 9: I don't typically think of the Barnett Shale as being in Northeast Texas – it appears more central than anything else.

Line 16: I'm not clear on the statement that TROPOMI is "very limited" on finer spatial scales. Does this mean TROPOMI can resolve one flux value for a 100km pixel and finer is not possible? Or is there some actual finer spatial threshold?

Line 17: 4-37% of total information. It is not clear what this means on reading the abstract at first, and even with the details later in the paper, it would be good to have further clarification on what this percentage is reported as relative to (what is "total information") in this sentence. This relates to clarifying what the 100 pieces of information is.

[Figure]

Line 20: Please be more specific here for the importance of 6ppb. My impression is there is an inflection point in performance at 6ppb where the resolved flux improves drastically.

Line 24: vague – please be more specific.

Line 51: Important to state the GHGsat performance is claimed but not proven.
* * *

---

## Author Comment (AC1) · 4 May 2018

**Response to Reviewer Comments:**

We thank the two Anonymous Reviewers for their comments.
* * *
**Reviewer #1 Comments:**

This paper describes results from an observing system simulation experiment (OSSE) to assess the potential to quantify sources of methane from different satellite observing systems. The focus is on emissions at the kilometre scale, both constant in time and transient. The paper is well written, and I recommend publishing after the following minor comments are addressed.

**General Comments:**

It is amazing to see the enormous improvement in capabilities over the past decades (and in the upcoming decade) to quantify emissions from space with ever increasing spatial (and temporal) resolution. However, one might ask, where the limit would be e.g. in spatial resolution to retrieve useful information e.g. in the context of mitigation. Given the focus on the kilometre scale, and on temporal variations down to hourly, why not go to even smaller scales? I suggest this should be discussed in the introduction to better motivate the targeted spatial and temporal scale.

Excellent question.  There were three primary reasons we chose to focus on the kilometer scale:
1) **Computational expense.**  Computational expense was a major factor in the choice of spatio-temporal resolution; it was a non-trivial endeavor to construct the footprints for this application.
2) **Availability of inventories.**  To our knowledge, there are not any methane inventories available at sub-kilometer scale that we could use to inform our analysis.
3) **Spatial resolution of current and future satellite-based instruments.**   The resolution chosen here is finer than the present satellite-based instruments (e.g., GOSAT, TROPOMI, and GeoCARB), so it seemed appropriate for this particular application.

There is ongoing work from a member in the Jacob group examining finer spatial scales than this (50m resolution), however this work is not yet published.

We have added the following text to the introduction:

Lines 66-68: "Our choice of scales is guided by the resolution of the planned satellite observations, and our choice of the Barnett Shale is guided by the availability of a high-resolution emission inventory for the region (Lyon *et al.,* 2015)."

And the following line in Section 2:

Lines 114-116: "Computing these high-resolution footprints was a non-trivial computational task and ultimately yielded more than 4 Tb of footprints for the week of pseudo-satellite observations in the Barnett Shale."

**Specific Comments:**

**1.)** Fig. 4: The methane enhancement looks somewhat patchy, with a number of white pixels with zero or near-zero enhancement next to pixels with significantly larger enhancements. Given that the atmosphere due to advection and mass conservation is expected to be continuous in those enhancements, and given that the grey-scale used for the visualization does not have any step changes, the figure is surprising.

The "patchy-ness" is actually due to a lack of data in a handful of locations. Constructing the footprints at this spatial scale required running 100 trajectories from 12 vertical layers for each column observation. We ultimately constructed more than 300,000 column observations which meant we had in excess of 3.6 million receptor locations for WRF-STILT (each with 100 particles). We included a number of fault tolerances but some of the simulations still crashed (e.g., due to reaching the wall clock for that particular job submission). If any of the 12 vertical layers failed to run successfully then we would have to throw out that column observation.

We have amended the caption to indicate that the patchy-ness is missing data, not zeros.

**2.)** Fig. 6, right panel: the colour regions don't follow the lines as they should (and as they do in the left panel).

This actually is correct. In the left panel of Fig. 6 (and 8) we vary the instrument precision and there is a monotonic response. In the right panel of Fig. 6 (and 8) we increase the number of return times and we find that the response is not actually monotonic. Black lines in the panel are the actual eigenvalues for each case and you'll notice that there are slight overlaps (or crossings) of the black lines. This is because there are a number of ways to change the return time for a satellite.

For example, for the daily observations we use data from 13 local time while the twice daily observations are at 10 and 14 (see Table 1). So the twice daily observations do not include the same observations as the daily observations. This means that the twice daily observations will not necessarily out-perform the daily observations (e.g., if there were more favorable meteorological conditions at 13 local time).

Regarding the shading in Fig. 6 (and 8), we tried a few different ways of presenting the results (e.g., coloring the individual lines) but it was quite messy because a number of the lines are quite close together. This seemed like the best way to present the results.

We have added the following text to the figure caption:

Fig 6 caption: "The change in flux threshold as the sampling frequency increases in the right panel is not necessarily monotonic, this is because some of the cases use different subsets

of observation (e.g., daily observations are at 13 local time while twice daily are at 10 and 14).”

**3.)** L334-336: Please clarify: you state “Analysis of HˆHˆT does not yield the eigenvectors of F”, but the previous sentence states otherwise.

Analysis of $\mathbf{\hat{H}}^T\mathbf{\hat{H}}$ and $\mathbf{\hat{H}}\mathbf{\hat{H}}^T$ yield the same eigenvalues but different singular vectors. This can be seen from a singular value decomposition of $\mathbf{\hat{H}}$ ($\mathbf{\hat{H}} = \mathbf{U\Sigma V}^T$; where $\mathbf{U}$ and $\mathbf{V}$ are unitary matrices: $\mathbf{I} = \mathbf{U}^T\mathbf{U} = \mathbf{UU}^T = \mathbf{V}^T\mathbf{V} = \mathbf{VV}^T$):

$$\begin{aligned}
\mathbf{\hat{H}}^T\mathbf{\hat{H}} &= (\mathbf{U\Sigma V}^T)^T\,\mathbf{U\Sigma V}^T \\
&= \mathbf{V\Sigma}^T\mathbf{U}^T\,\mathbf{U\Sigma V}^T \\
&= \mathbf{V\Sigma}^T\mathbf{\Sigma V}^T \\
&= \mathbf{V\Lambda V}^T
\end{aligned}$$

$$\begin{aligned}
\mathbf{\hat{H}}\mathbf{\hat{H}}^T &= \mathbf{U\Sigma V}^T(\mathbf{U\Sigma V}^T)^T \\
&= \mathbf{U\Sigma V}^T\,\mathbf{V\Sigma}^T\mathbf{U}^T \\
&= \mathbf{U\Sigma\Sigma}^T\mathbf{U}^T \\
&= \mathbf{U\Lambda U}^T
\end{aligned}$$

From this, we can see that analysis of $\mathbf{\hat{H}}^T\mathbf{\hat{H}}$ and $\mathbf{\hat{H}}\mathbf{\hat{H}}^T$ would yield the same singular values ($\mathbf{\Lambda}$), that can be related back to the eigenvalues, but different singular vectors. This means that, depending on the dimension of $m$ and $n$, we can obtain the eigenvalues by analyzing either $\mathbf{\hat{H}}^T\mathbf{\hat{H}}$ or $\mathbf{\hat{H}}\mathbf{\hat{H}}^T$.

We have updated the text in the appendix.
* * *
**Reviewer #2 Comments:**

Turner et al. present an OSSE to assess the performance of different space-based methane measurements (TROPOMI, GeoCarb, aspirational), in particular considering the ability of these different sensors to evaluate methane emissions from the Barnett Shale, a major oil and gas production region in US. This manuscript is very well written – clear, concise, and presents interesting results of particular relevance at this junction in time. I'm supportive of publication once my minor concerns mentioned below are addressed.

**General Comments:**

**1.)** Larger context issue: The work as presented lacks some context that limits the extent and value of the conclusions. This could be addressed easily and would make the assessment of what the Barnett shale region looks like as a source region compared to other regions and sources of methane. Is the Barnett a typical oil/gas field (for the US, for the globe)? Are emissions particularly large (or small) from this region? Are emissions particularly spatially heterogeneous (lot of intense point sources? Heavy- tail distribution of emissions?)? How does this compare to other interesting methane source regions? Would

results be extensible to other oil/gas regions? To regions with intense wetlands? The work presented in convincing for the capabilities/limitations of different sensors but I don't know if the 6ppb suggested observational threshold is actually an important threshold for studying any domain other than the Barnett.

Much of the information requested is not available for other regions. To our knowledge, the Barnett Shale is the only oil/gas basin with a high-resolution inventory available (the inventory constructed by the EDF). So it is not easy to compare the distribution of sources to another basin. The availability of a detailed inventory was a major motivator in the choice of the Barnett Shale for this OSSE.

We have added the following text to the introduction:

Lines 68-70: "The pattern and density of methane emissions in the Barnett Shale is typical of other source regions in the US (Maasakkers *et al.*, 2016)."

**2.)** Question on methodology: What is the impact of choosing to only simulate observations made within the region defined (dashed orange box in Fig. 2)? All the sensors considered would make observation surrounding this box as well, which would have overlapping sensitivity with this region. How does neglecting these observations impact the results? In particular, for sensors like TROPOMI with 'coarser' resolution, might the use of these observation points actually improve the results?

Excellent question. Our present study limited the observation domain to the dashed orange box due to computational expense. Constructing the footprints at the fine spatial scales here required running 100 trajectories from 12 vertical layers for each column observation. We ultimately constructed more than 300,000 column observations, which meant we had in excess of 3.6 million receptor locations for WRF-STILT (each with 100 particles). The library of footprints for this dashed orange box is more than 4 Tb.

However, my previous work has addressed what amounts to effectively the same question just phrased slightly different: "what is the impact of limiting the domain". This previous work (Turner et al., 2016; Supplemental Section 6.1) analyzed the impact of domain size on the error reduction for WRF-STILT inversions in California's Bay Area. We found that it made little difference in that application. That study used "error reduction" as the metric and found roughly 1% less error reduction when using the reduced domain, compared to the base case.

Further, the total weekly footprint (bottom right panel of Fig. 2) shows that footprints are strongly sensitive to the core 70×70 km$^2$ region.

We have added the following text:

Lines 121-129: "Additional observations within the outer domain would need to be considered to constrain emissions in that domain. On the other hand, information on emissions in the 70×70 km$^2$ core domain is mainly contributed by observations within the domain. Thus our focus will be to determine the capability of the observations in the 70×70 km$^2$ domain to constrain emissions within that same domain, but we include the outer

290×235 km$^2$ domain in our footprint analysis for completeness in accounting of information. Previous work from Turner et al. (2016; Supplemental Section 6.1) investigated the impact of domain size on error reduction for WRF-STILT inversions in California's Bay Area and found that it had a negligible impact."

**Minor Comments:**

**1.)** Line 9: I don't typically think of the Barnett Shale as being in Northeast Texas – it appears more central than anything else.

We have updated the text to refer to it as "Barnett Shale region in Texas".

**2.)** Line 16: I'm not clear on the statement that TROPOMI is "very limited" on finer spatial scales. Does this mean TROPOMI can resolve one flux value for a 100km pixel and finer is not possible? Or is there some actual finer spatial threshold?

**3.)** Line 17: 4-37% of total information. It is not clear what this means on reading the abstract at first, and even with the details later in the paper, it would be good to have further clarification on what this percentage is reported as relative to (what is "total in- formation") in this sentence. This relates to clarifying what the 100 pieces of information is.

**4.)** Line 20: Please be more specific here for the importance of 6ppb. My impression is there is an inflection point in performance at 6ppb where the resolved flux improves drastically.

**5.)** Line 24: vague – please be more specific.

We have amended the abstract in response to points 2-5.

Lines 15-16: "We find that a week of TROPOMI observations should provide information on temporally invariant emissions at ~30 km spatial resolution."

Lines 16-18: "GeoCARB should provide information available on temporally invariant emissions ~2-7 km spatial resolution depending on sampling frequency (hourly to daily)."

Lines 19-20: "A precision better than 6 ppb is critical for GeoCARB to achieve fine resolution of emissions."

Further discussion of these points was also added to the results section (see tracked changes on Pages 11-12).

**6.)** Line 51: Important to state the GHGsat performance is claimed but not proven.

We no longer mention GHGSat as it is not really relevant to the discussion here.